# Polar Solomon rings in ferroelectric nanocrystals

Jing Wang [1,2,9], Deshan Liang [1,9], Jing Ma [2,9], Yuanyuan Fan[1], Ji Ma[2,3], Hasnain Mehdi Jafri[1], Huayu Yang[1], Qinghua Zhang[4], Yue Wang[2], Changqing Guo [1], Shouzhe Dong[1], Di Liu[1], Xueyun Wang [5], Jiawang Hong [5], Nan Zhang[6], Lin Gu[2,4], Di Yi [2], Jinxing Zhang[7], Yuanhua Lin [2], Long-Qing Chen [8], Houbing Huang [1] ✉ & Ce-Wen Nan [2] ✉

Solomon rings, upholding the symbol of wisdom with profound historical roots, were widely used as decorations in ancient architecture and clothing. However, it was only recently discovered that such topological structures can be formed by self-organization in biological/chemical molecules, liquid crystals, etc. Here, we report the observation of polar Solomon rings in a ferroelectric nanocrystal, which consist of two intertwined vortices and are mathematically equivalent to a $4_1^2$ link in topology. By combining piezoresponse force microscopy observations and phase-field simulations, we demonstrate the reversible switching between polar Solomon rings and vertex textures by an electric field. The two types of topological polar textures exhibit distinct absorption of terahertz infrared waves, which can be exploited in infrared displays with a nanoscale resolution. Our study establishes, both experimentally and computationally, the existence and electrical manipulation of polar Solomon rings, a new form of topological polar structures that may provide a simple way for fast, robust, and high-resolution optoelectronic devices.

Solomon links/rings (Fig. 1a), named after King Solomon[1], hold the shape of four crossings and comprise two components (red and blue rings), and are mathematically described as a $4_1^2$ link, one of the topological structures in mathematic knot and link theory. Knots and links, composed of closed ring/rings in three-dimensional (3D) space in mathematics, are classified by the numbers of their crossings and components of rings[2] (Supplementary Fig. 1). Such knots and links-based topological structures have been widely observed in nature and/ or artificially constructed, e.g., Trefoil knots, Hopf links, Solomon rings, and Borromean rings, in biological/chemical molecules[3–12], liquid crystals[13–15], quantum matters[16–20], etc.

Very recently, a polar Hopfion has been computationally predicted in a $PbZr_{0.6}Ti_{0.4}O_3$ nanocrystal[21], which is totally different from the previous reported topological polar textures[22–27], indicating the feasibility of fabricating knots and links in complex oxides with ferroelectric order parameters. Here, we consider a 3D vortex structure as a polar ring from the perspective of polar conservation, e.g., $\sum p_i = 0 (i = x, y, z)$, where $p_i$ is polarization components along

[1]Advanced Research Institute of Multidisciplinary Science, and School of Materials Science and Engineering, Beijing Institute of Technology, 100081 Beijing, China. [2]State Key Laboratory of New Ceramics and Fine Processing, School of Materials Science and Engineering, Tsinghua University, 100084 Beijing, China. [3]School of Material Science and Engineering, Kunming University of Science and Technology, 650093 Kunming, Yunnan, China. [4]Beijing National Laboratory for Condensed Matter Physics, Institute of Physics, Chinese Academy of Science, 100190 Beijing, China. [5]School of Aerospace Engineering, Beijing Institute of Technology, 100081 Beijing, China. [6]Beijing Engineering Research Center of Mixed Reality and Advanced Display, and School of Optics and Photonics, Beijing Institute of Technology, 100081 Beijing, China. [7]Department of Physics, and Key Laboratory of Multi-scale Spin Physics, Ministry of Education, Beijing Normal University, 100875 Beijing, China. [8]Department of Materials Science and Engineering, Pennsylvania State University, University Park, PA 16802, USA. [9]These authors contributed equally: Jing Wang, Deshan Liang, Jing Ma. ✉e-mail: hbhuang@bit.edu.cn; cwnan@tsinghua.edu.cn

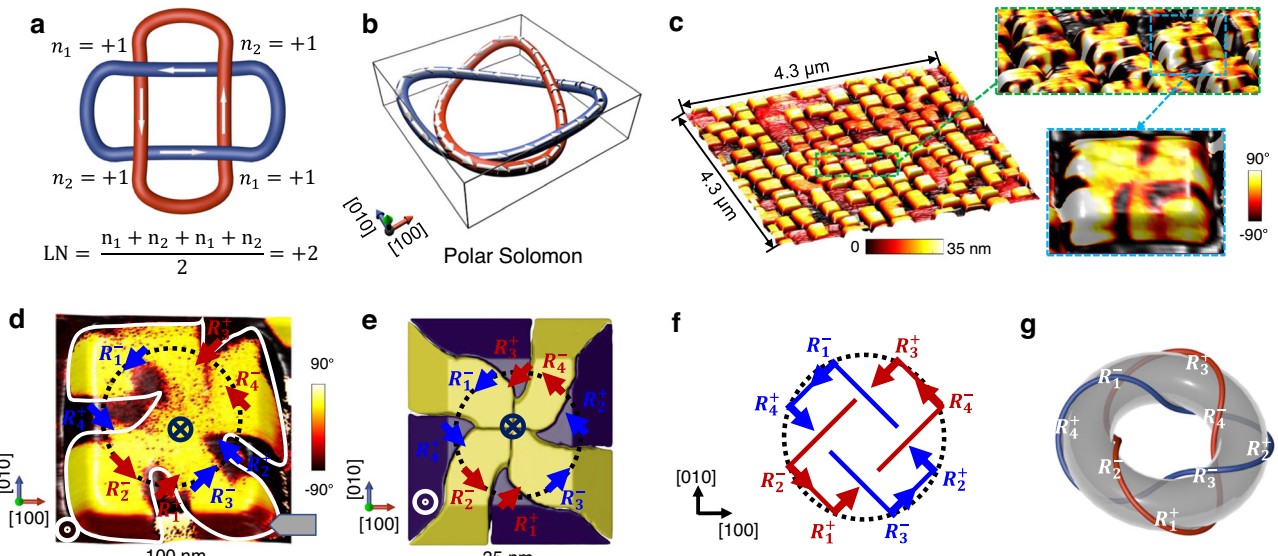

**Fig. 1 | Polar Solomon rings. a** The definition of polar Solomon rings with the linking number (LN) of +2, from the mathematical point of view. **b** Distorted 3D polar Solomon rings, which can be considered as two intersected vortices composed of eight polarization variants from BiFeO₃ as shown in Fig. S2. **c** A large-scale BiFeO₃ nanocrystal array (morphology, left) with the magnified OOP domain patterns (right). The lateral size and height of a nanocrystal are about 300 nm and 35 nm, respectively. **d** The construction of OOP and IP polarization projection for a BiFeO₃ nanocrystal by PFM measurement. The profile of the downward domain pattern is outlined by the white line to guide one's eyes. **e** Phase-field simulations of the domain pattern for BiFeO₃ nanocrystal. **f** The extracted two intersecting polar vortices projected in (001)-plane. **g** The polar Solomon rings wrapped in a donut extracted from **e**.

Cartesian coordinates (Supplementary Fig. 2). This ring is the basic component for the construction of topological polar Solomon based on the knot and link theory (Fig. 1b). Based on this viewpoint, we demonstrate the natural formation of polar Solomon rings in the well-known rhombohedral ferroelectric BiFeO₃ nanocrystals[28].

## Results

### Observation of polar Solomon rings in BiFeO₃ nanocrystals

We explored the polarization distribution in BiFeO₃ nanocrystal array by the combination of piezoresponse force microscopy (PFM) and transmission electron microscopy (TEM). The detailed growth process for the nanocrystals self-assembled on (001)-oriented LaAlO₃ substrate can be seen in Methods and the previous report[29]. A large-scale BiFeO₃ nanocrystal array with the gradually magnified out-of-plane (OOP) domain pattern is shown in Fig. 1c. To understand the detailed domain structure, we investigate the OOP (Fig. 1d) and in-plane (IP) (Supplementary Fig. 3) polarization projection in one BiFeO₃ nanocrystal. The OOP PFM phase image indicates the alternating downward (⊗) and upward (⊙) polarization projection along [001] crystalline orientation, consistent with the coexistence of upward and downward domain variants in the cross-sectional TEM image (Supplementary Fig. 4). The corresponding strain distribution of the cross-sectional nanocrystal can be seen in Supplementary Fig. 5. The IP polarization projection (indicated by blue and red arrows in Fig. 1d) is further confirmed by multiple IP PFM phase images with the sample rotated by different angles with respect to the cantilever.

For further illustration, we also performed phase-field simulations to reconstruct the 3D domain pattern in the BiFeO₃ nanocrystal with a dimension of $93 \times 104 \times 20$ nm³. The detailed island conditions can be seen in supplementary Fig. 6. The simulated 3D domain structure (Fig. 1e) indicates that the spatial distribution of the domain order is consistent with our experimental observation in Fig. 1d. Intriguingly, by comprehensively considering the OOP and IP polarization projection of the specimen, the domain structure of the BiFeO₃ nanocrystal can be extracted as two intersected 3D polar vortices, one is composed of polarization variants with $R_4^-[1\bar{1}\bar{1}]$, $R_3^-[\bar{1}\bar{1}\bar{1}]$, $R_2^-[1\bar{1}\bar{1}]$, $R_1^+[111]$ (marked by the red ring in Fig. 1f), and the other one is composed of polarization

variants with $R_4^+[1\bar{1}1]$, $R_3^+[\bar{1}1\bar{1}]$, $R_2^+[\bar{1}11]$, $R_1^-[\overline{111}]$ (marked by the blue ring in Fig. 1f). The two polar rings can also be extracted as the polar Solomon rings across a donut, as shown in Fig. 1g and Supplementary Fig. 7. The detailed transformation process from the 3D domain in BiFeO₃ nanocrystal to the polar Solomon rings across a donut can be seen in Supplementary Text, Supplementary Fig. 8 and Supplementary Movie 1. We first identify the linking number (LN) of the observed polar Solomon rings. Since the polarization in the blue ring and the red ring can rotate either clockwise or counterclockwise, we considered all the possible interwoven ways for these two polar rings, as seen from the detailed illustration in Supplementary Text and Supplementary Fig. 9. We conclude that there are only two types of polar Solomon rings with the corresponding LN of +2 and −2, as demonstrated in Fig. 1 and Supplementary Figs. 10 and 11, respectively.

### Topological property for polar Solomon rings

To further confirm the topological characteristic of this polar texture, we characterize its toroidal moment and winding number based on results from phase-field simulations. Figure 2a shows the corresponding color domains with 3D configuration. We divided the nanocrystal into four layers L1, L2, L3, and L4, and the corresponding color domains are shown in Fig. 2b. We observe that each of the color domains twists from top to bottom, e.g., the four center-domains change from $R_1^-[\bar{1}\bar{1}1]$, $R_2^-[1\bar{1}\bar{1}]$, $R_3^-[\bar{1}\bar{1}\bar{1}]$ and $R_4^-[1\bar{1}\bar{1}]$, to $R_1^+[111]$, $R_2^+[\bar{1}11]$, $R_3^+[\bar{1}1\bar{1}]$ and $R_4^+[1\bar{1}1]$. To calculate the local toroidal moments, polar evolution along the circumferences with two different radii ($R1$ and $R2$ in Fig. 2a, c) is extracted from the polar vector image in Fig. 2c, d. The corresponding $x$, $y$, and $z$ components of the polarization along these two circles are plotted as a function of the polar angle of the polar coordinate, as shown in Fig. 2e. According to the equation for calculating the toroidal moment[30,31],

$$G = (2N)^{-1} \Sigma_i R_i \times p_i, (i = x, y, z) \quad (1)$$

the $x$, $y$, and $z$ components of the local toroidal moment are calculated as shown in Fig. 2f. The non-zero value of $G_z$ indicates the IP topological configuration of the polar Solomon rings. Furthermore,

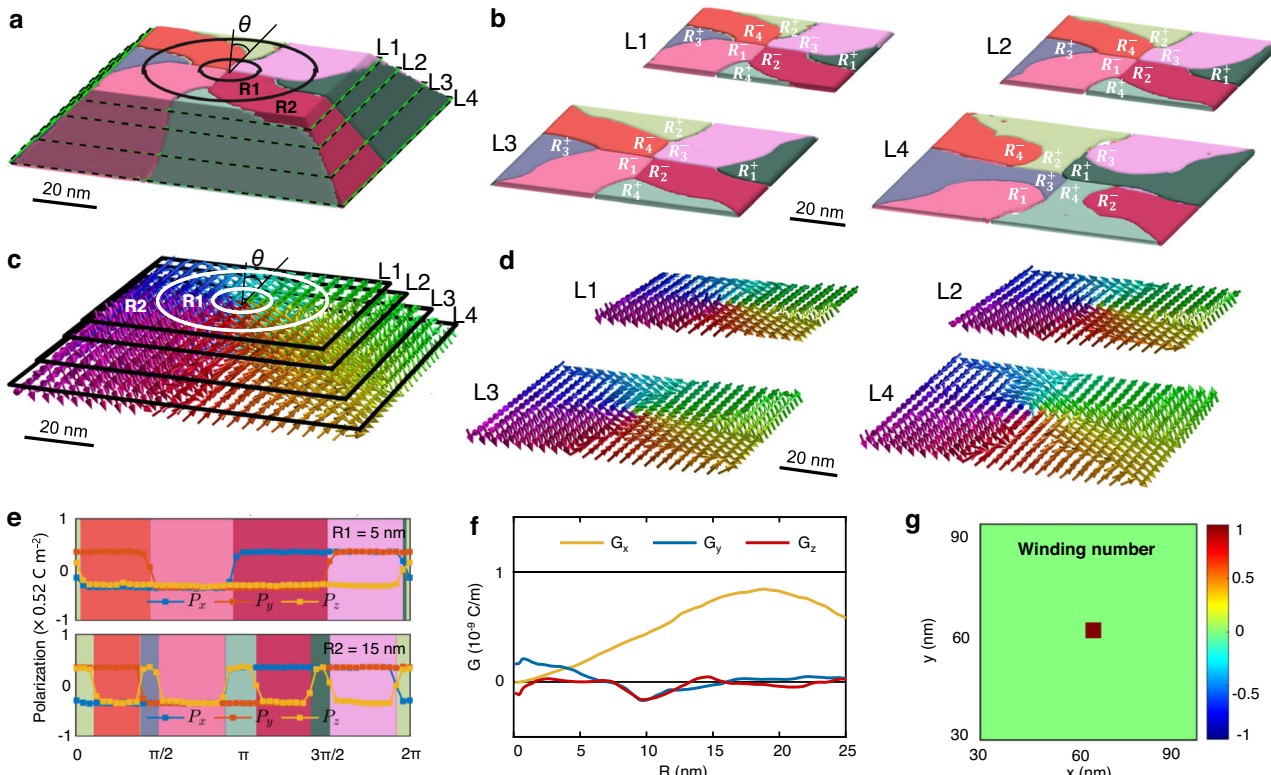

**Fig. 2 | Topological feature for the polar Solomon rings characterized by phase-field simulations. a** Phase-field simulations of the 3D domain pattern for the polar Solomon rings. **b** The domain pattern in L1, L2, L3, and L4 layers, respectively. **c, d** The corresponding polar vector mapping images for the data in **a** and **b. e** $P_x$-, $P_y$-, and $P_z$-component of each domain evolution in L1 as a function of the rotating angle ($\theta$) with the polar coordinates, where the cases of R1 = 5 nm and R2 = 15 nm are considered. **f** The calculated $x$, $y$, and $z$ components of the toroidal moment. **g** The calculated winding number for the polar vector map in L4 (D). The red dot in the image center indicates the winding number of 1 in the domain center and zero in the other locations.

we also calculated the winding number of L4 based on the polar vector map in Fig. 2d using the following equation[32]:

$$n = \frac{1}{2\pi} \int_0^{2\pi} \frac{d\phi}{d\theta}(r,\theta)d\theta \qquad (2)$$

where $\varnothing$ is the angle of the IP projection of the polar vector, $\theta$ is the polar angle of the polar coordinate, and $r$ is the integrating radius. The winding number of 1 in the center of Solomon rings and 0 in the other spaces (Fig. 2g) indicates a polar vortex in the center. The topological characteristic and the mutual embracing of the two vortices can also be deduced from the normal strain ($\varepsilon_{xx}$, $\varepsilon_{yy}$, $\varepsilon_{zz}$) and shear strain ($\gamma_{yz}$, $\gamma_{xz}$, $\gamma_{xy}$) (Supplementary Fig. 12).

Several factors are discussed for the formation of polar Solomon rings. First, the epitaxial strain and the flexoelectric effect are negligible. The epitaxial strain is relaxed by the arrayed dislocations at the film/substrate interface, as demonstrated by the GPA analysis for the HAADF image in our previous work[33]. Normally, the flexoelectric effect operates at several nanometer scales[34], so it has a negligible effect on the formation of Solomon rings due to the large size of the nano-island. The depolarization field, rhombohedral ferroelectric phase and nanocrystal geometry are important for the formation of this polar structure. From phase-field simulations, we conclude that it is necessary to consider the depolarization field for the construction of polar Solomon rings (Supplementary Fig. 13). According to the knots and links theory (Fig. 1a and Supplementary Fig. 1), the polar Solomon rings can be formed in the rhombohedral ferroelectric phase because it has 8 variants and could form two crossed vortices in 3D space with four crossing points, which is the same as the mathematical definition of a $4_1^2$ link. The rectangular shape of the nano-island constrains the eight

domains and makes the two polar rings twisted and connected as a way of Solomon rings.

## Electric field-driven topological phase transition

The topological polar Solomon rings are observed in all other BiFeO₃ nanocrystals, as demonstrated in Fig. 3, Supplementary Figs. 14 and 15. To further explore the reconfigurable feature of the polar Solomon rings, we investigate the domain evolution of one BiFeO₃ nanocrystal under an external electric field. As shown in the OOP and IP PFM phase images in Fig. 3a, b, the polar texture in BiFeO₃ nanocrystal exhibits Solomon rings, as illustrated by the interwoven polar rings drawn with red and blue arrows. With further increase in the applied electric voltage to −4 V, the red ring and the blue ring are interrupted and a vertex topological structure is emerging, i.e., the four-fold quad-domains with a center-divergent polar configuration. Intriguingly, when a reversed electric voltage of 2 V is applied to the scanning probe, this vertex structure is transformed to the Solomon rings with eight alternating upward and downward domains again. When the reversed electric voltage further increases to 3 V, the Solomon rings then transform to the topological vertex structure with center-convergent polar configuration. When a reversed small upward field (−2 V) is applied, the vertex structure is transformed to Solomon rings again. We also show the raw data in supplementary Fig. 16. Our observation demonstrates that the topological phase transition between polar Solomon rings and polar vertex structure can be deterministically and reversibly controlled in BiFeO₃ nanocrystals by external electric fields. The reversible regulation of Solomon rings and vertex quad-domains is also demonstrated by phase-field simulations (Fig. 3c), which shows the similar domain evolution with the experimental results. The above domain transformation is further confirmed by the line-profile analysis of the sequence of OOP

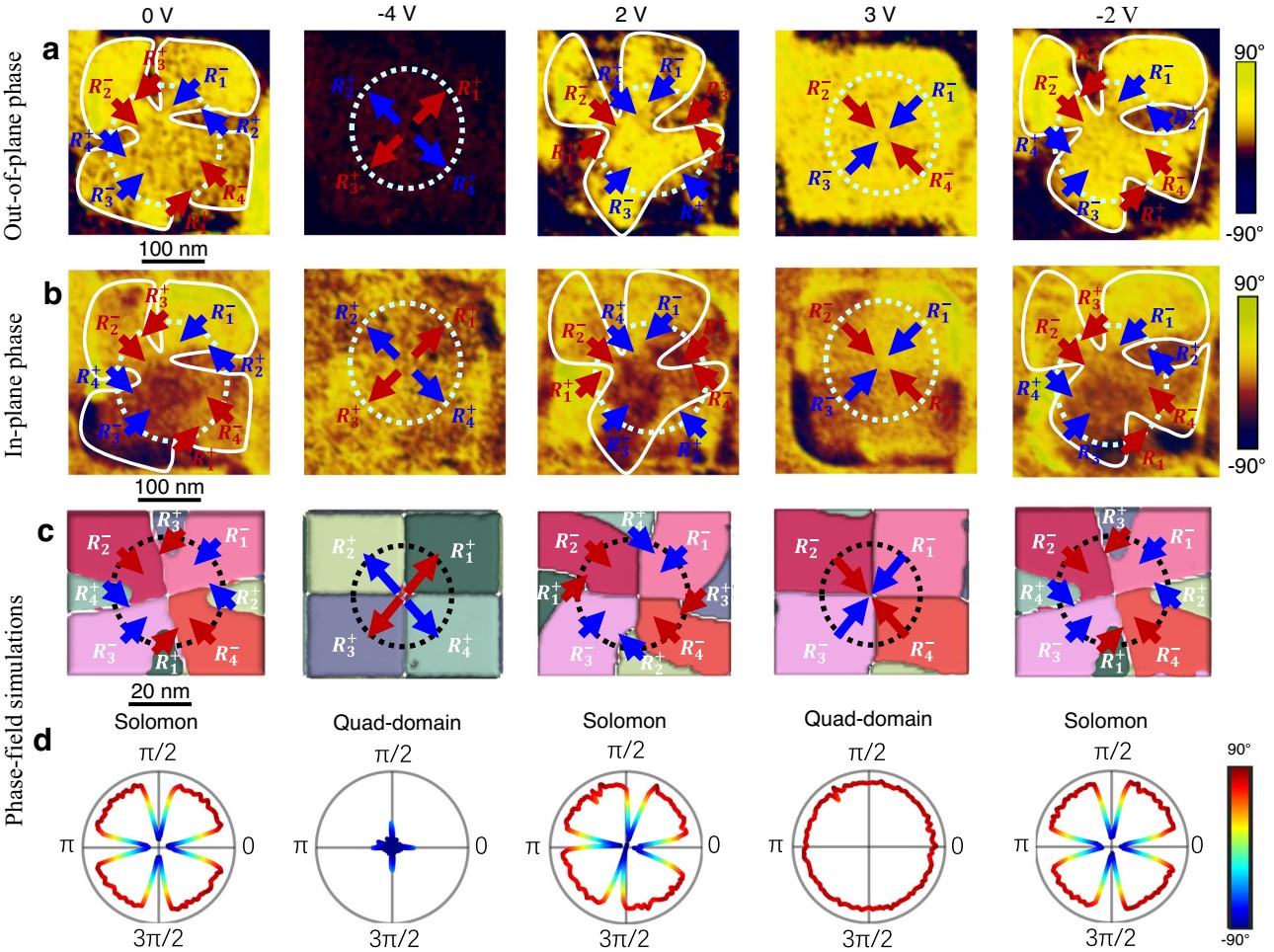

**Fig. 3 | Electric-field-driven topological phase transition between polar Solomon rings and vertex structures in BiFeO$_3$ nanocrystals. a**, **b** OOP (**a**) and IP (**b**) PFM phase images for the domain pattern in BiFeO$_3$ nanocrystal when the electric fields of 0 V, −4 V, 2 V, 3 V, and −2 V are applied to the scanning probe, respectively. **c** Phase-field simulations for the domain evolution in BiFeO$_3$ nanocrystal when exposed to different poling voltages. **d** The corresponding line profile for the OOP polarization projection along the dashed circular loop in **c**.

polarization projection along the circles in Fig. 3c with polar coordinate, as shown in Fig. 3d (simulation results) and Supplementary Fig. 17 (experimental results). The flower-like patterns in Fig. 3d and Supplementary Fig. 17 indicates the alternative upward and downward domain along the circles in the Solomon rings, while the dot- and circle-like patterns indicate either pure upward domain or downward domain for center-divergent and center-convergent vertex quad-domains, respectively. The dynamic domain evolution between the Solomon rings and vertex-like quad-domains under an electric field can also be seen in Supplementary Movie 2 and Supplementary Movie 3.

**Infrared absorption for various topological structures**
There have been a number of recent reports on the interactions between ferroelectric polar structures and ultrafast light, including anisotropic absorption of the THz wave in ferroelectric polymer within different sub-domains[31], collective dynamics for the vortexon in PbTiO$_3$/SrTiO$_3$ superlattices[35], ultrafast driven nanocrystal[36], as well as the domain-wall ultrafast dynamics[37]. Here, we explore the possibility of THz absorption behavior in the BiFeO$_3$ nanocrystals with multiple topological polar textures and a much higher curie temperature (~1103 K). We first performed the Fourier-transform infrared spectroscopy (FTIR) for R-phase BiFeO$_3$ thin film (Supplementary Fig. 18). There is a typical absorption at ~500 cm$^{-1}$ for the specimen. This absorption peak arises from the vibration of O-Fe-O bond[38]. To further explore the IR absorption for different polar topological textures in

BiFeO$_3$ nanocrystals, we performed atomic force microscope-infrared spectroscopy (AFM-IR) with a spatial resolution of ~10 nm (Fig. 4a). Excitingly, we observed that the different topological polar structure shows distinct IR absorption intensity but with the similar wave number (~1100 cm$^{-1}$), which is consistent with the doubling frequency absorption compared with the FTIR result. As shown in the red and blue curve in Fig. 4b, the IR absorption is strong in upward and downward quad-domains (U-Q and D-Q), while it is weak in the Solomon rings (see green curve in Fig. 4b). This result is further demonstrated by the OOP PFM (Fig. 4c–e) and AFM-IR mapping (Fig. 4f–h). The cross-sectional line-profile (Fig. 4f–h) indicates the absorption is strong for the nanocrystal with U-Q and D-Q, while it is the weakest for the Solomon rings.

The absorption of the THz electromagnetic wave is related to the relative angle between the ferroelectric polarization and the electromagnetic wave polarization. If they are parallel or antiparallel, the absorption is strongest, but if they are perpendicular to each other, the absorption is weakest[31]. In addition, domain walls are inactive in the THz band due to their long relaxation time, and they act as defective interfaces arising from their frustrated dipoles. These unfulfilled dipoles in the domain walls contribute less to the permittivity than their counterparts in the domains[39]. Therefore, if the volume fraction of the domain walls is large, the absorption intensity of the THz electromagnetic wave would decrease. The electromagnetic wave we used is out-of-plane polarization, see supplementary Fig. 19a. For the polar structures, e.g., Solomon rings (supplementary Fig. 19b) and quad-

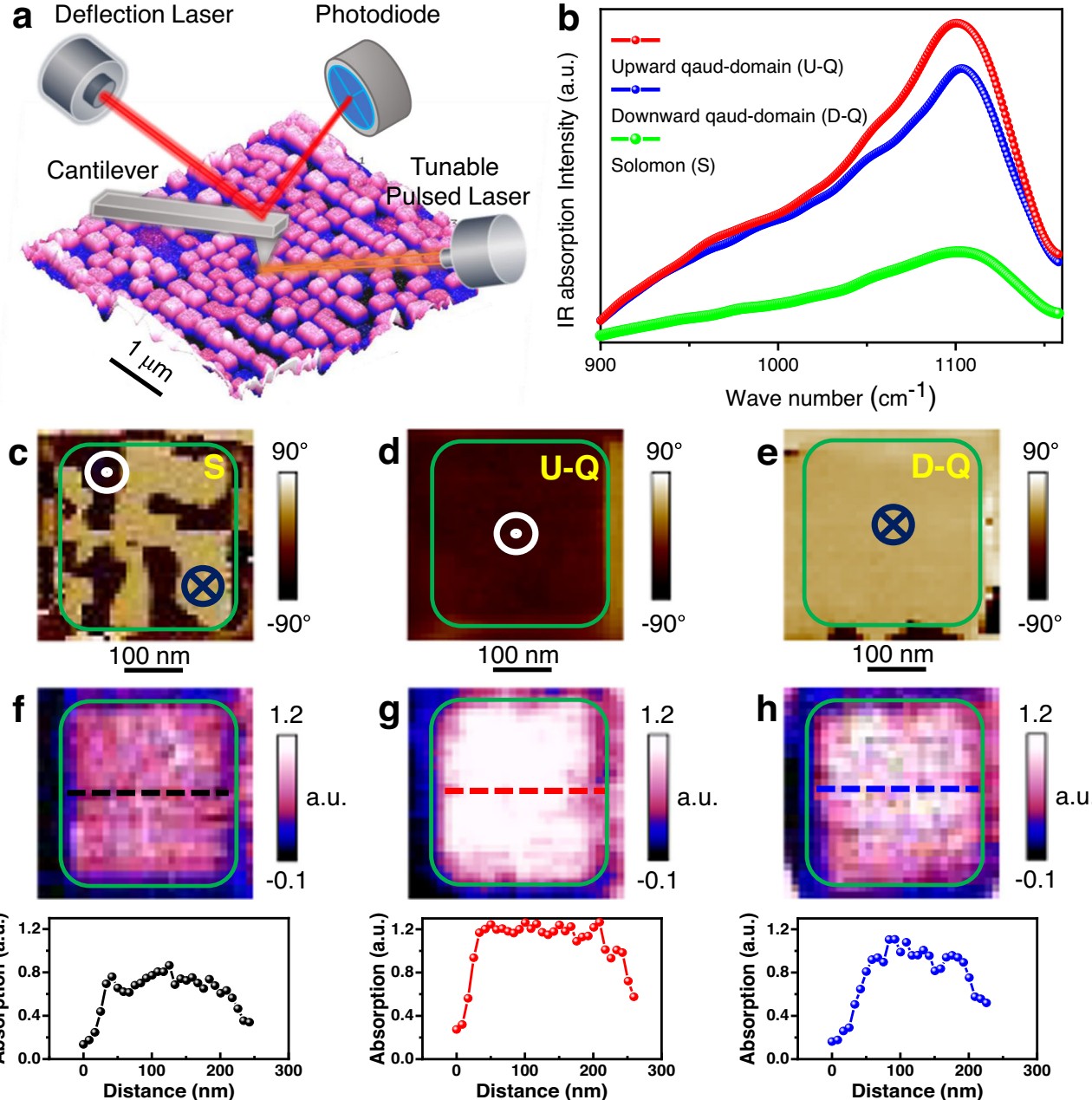

**Fig. 4 | Distinct IR absorption for BiFeO₃ nanocrystals with different topological structures. a** Schematic of experimental setup for IR absorption of BiFeO₃ nanocrystals. **b** The different IR absorption spectrum for upward quad-domains (U-Q), downward quad-domains (D-Q) and Solomon rings (S). **c–e** OOP phase image for **c** S, **d** U-Q, and **e** D-Q. **f–h** IR absorption image and the corresponding line profile for **c** S, **d** U-Q, and **e** D-Q.

domains (Supplementary Fig. 19c, d), their out-of-plane polarization projection is either parallel or antiparallel to the electromagnetic polarization, but the main difference is the volume fraction of the domain walls. For the quad-domains, there are 4 domain variants and the volume fraction of the domain walls is small. However, for Solomon rings, the eight-domain variants are twisted in the 3D space, inducing a larger volume fraction of the domain walls than the quad-domains. Thus, we observe a weaker THz wave absorption in Solomon rings than in the quad-domains. This finding is further demonstrated by the homogeneous IR absorption in continuous BiFeO₃ thin films with upward and downward domain structures (Supplementary Fig. 20). Furthermore, we also compared the IR absorption by using the THz wave with OOP and IP electric field vector (Supplementary Fig. 21), which shows the similar results, consistent with the 3D domain of polar Solomon rings with both OOP and IP polarization projection.

## Discussion

Benefiting from the different IR absorption for quad-domains and Solomon rings, we also demonstrate the reliable control of the IR response in BiFeO₃ nanocrystals. A large area of BiFeO₃ nanocrystals with polar Solomon rings is selected as the weak IR absorption matrix, as shown in Fig. 5a–c. To create alternative vertex quad-domains and Solomon rings, an electric voltage is alternatively applied to the adjective nanocrystals with initial topological Solomon rings (Fig. 5d), and the IR response changes from the uniform absorption to the alternative strong and weak absorption (Fig. 5e, f). With the BiFeO₃ nanocrystal array, the above controllable topological domain structures with distinct IR response may pave the way for the design of IR display (Fig. 5g). To demonstrate the display function of those topological polar nanocrystals, we poled several selected nanocrystals containing the quad-domain state with upward polarization and to

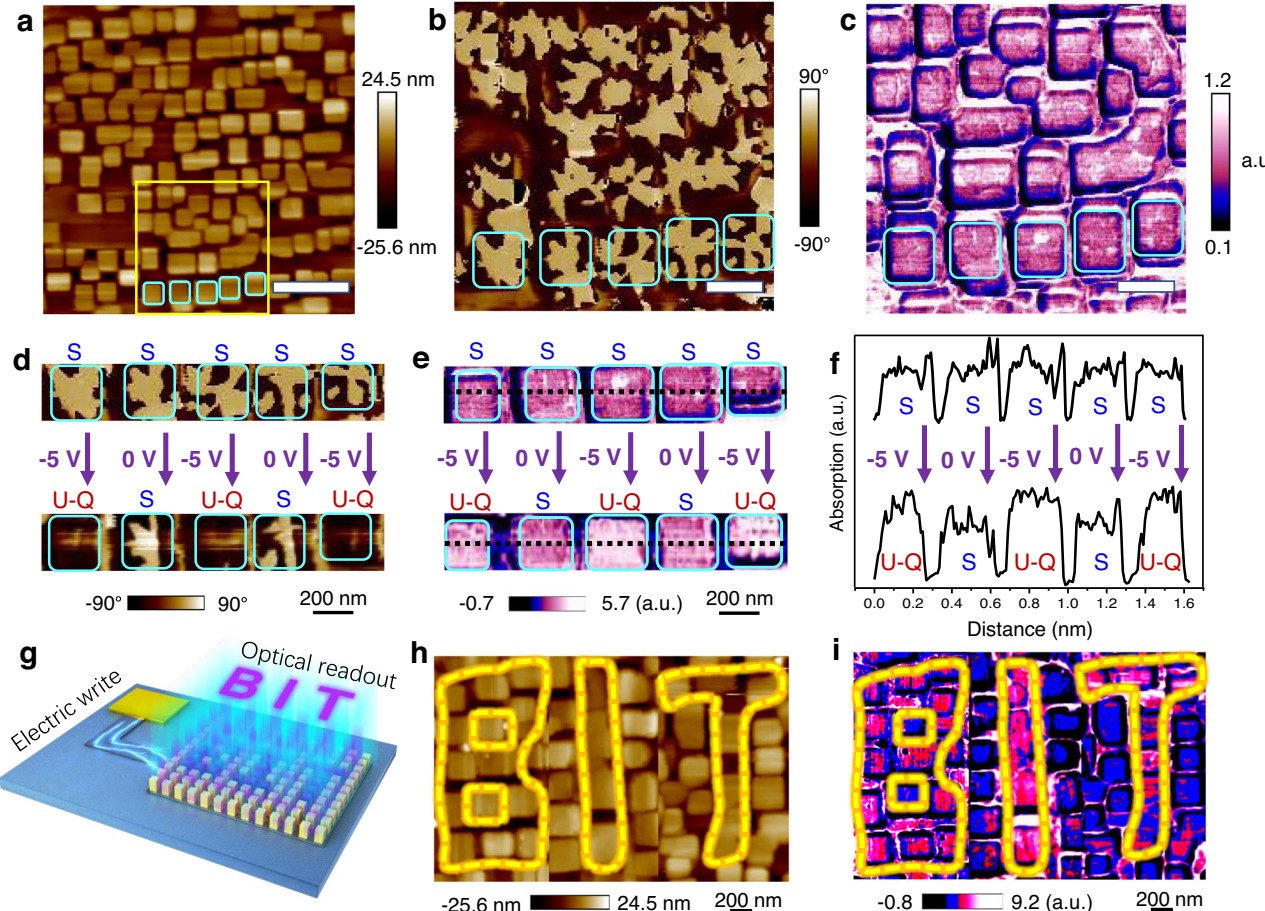

**Fig. 5 | The demo of IR display by electric poling the selected nanocrystals.**
**a** Morphology, **b** OOP PFM phase image and **c**, IR absorption image for a large-scale BiFeO₃ nanocrystal array with the Solomon rings (S). The scale bar in **a–c** is 1 μm, 0.3 μm, and 0.3 μm, respectively. **d** OOP phase image and **e** IR absorption image for BiFeO₃ nanocrystals with S structure (up panel), and alternative U-Q and S structures (bottom panel). **f** IR absorption line profile along the black dashed line in e. **g** Schematic diagram showing the IR display by electric control of the domain pattern in the nanocrystal array. **h** Electric writing 'BIT' characters by poling the corresponding nanocrystals to U-Q structure (outlined by closed yellow loop), while the others maintain S structure. **i** IR absorption image showing 'BIT' characters in the electric written nanocrystal array.

form a 'BIT' symbol in the Solomon matrix (Fig. 5h), and the AFM-IR image shows the strong IR absorption in 'BIT' but weak IR absorption in the other nanocrystals (Fig. 5i). The IP PFM phase image after poling can be seen in supplementary Fig. 22. One thing should be noted is that the rhombohedral BiFeO₃ nano-islands are embedded in the tetragonal BiFeO₃ thin films, and according to the FTIR spectrum (Supplementary Fig. 18), the tetragonal BiFeO₃ thin-film shows higher absorption intensity than the rhombohedral BiFeO₃ thin-film around 550 cm⁻¹, which explains the higher absorption intensity (white) in the regions between nanocrystals than that of unpoled nanocrystals. The above observation may demonstrate the potential applications of these topological polar textures in imaging, display, night vision, etc. It is worth mentioning that, the relaxation time of the topological phase transition between vertex quad-domains and Solomon rings depends on the ferroelectric polarization switching, e.g., a time constant of sub-nanoseconds[40], indicating the ultrafast conversion speed for the displayed images. Another advantage of this ferroelectric nanocrystal display is the resolution due to the size of the nanocrystals, which is 200–300 nm. Each nanocrystal can serve as an addressable pixel, which is much smaller than the pixel size (several to hundreds μm) of the other display objects[41].

In summary, we discover the polar Solomon rings in BiFeO₃ nanocrystals using a combination of theoretical analysis, phase-field simulations, and experimental observations. They are composed of two interwoven vortices in 3D and are equivalent to a $4_1^2$ link in

mathematics with a topological winding number of 1. We demonstrate the electric-field manipulated topological phase transition in the BiFeO₃ nanocrystals between the Solomon rings and the vertex with four-fold quad-domains, as well as the distinct IR responses and the potential photoelectronic applications. The polar Solomon rings in the BiFeO₃ nanocrystals reported here represent an example where the topologically non-trivial polar entities are defined based on knots and links theory. This provides a glimpse of the large variety of topological polar rings that may be expected from other ferroelectric nanocrystals with various crystalline symmetries, and the potential photoelectric devices in the terahertz range.

## Methods

### Nanocrystals preparation
BiFeO₃ nanocrystals were deposited on (001)-oriented LaAlO₃ substrate by the pulsed laser deposition technique. The detailed mechanism of nano-island growth mode was clarified in our previous work[28], where (La,Sr)MnO₃ wetting layer and growth temperature are two important parameters. In this work, to stimulate the island growth mode for BiFeO₃ nanocrystals, ~2 nm thick (La,Sr)MnO₃ buffer layer was pre-grown on LaAlO₃ substrate to reduce the surface energy of (La,Sr)MnO₃/LaAlO₃ system. And a growth temperature of 700 °C is used to induce interfacial dislocations to release the epitaxial strain and nucleate rhombohedral-phase nano-islands. The metallic phase (La,Sr)MnO₃ can also serve as the bottom electrode for switching ferroelectric

polarization. During $BiFeO_3$ and $(La,Sr)MnO_3$ thin-film growth, a KrF excimer laser with a wavelength of 248 nm was used, and the corresponding repetition rate and energy density of the laser were 5 Hz and ~1.5 J/cm², respectively. The thin films were grown under an oxygen pressure of 0.2 mbar. After the deposition, the thin films were slowly cooled down to room temperature under 200 mbar oxygen pressure.

### PFM measurement

The PFM measurements were performed at room temperature by using an Infinity Asylum Research AFM and a Bruker Icon AFM. To construct the 3D polarization distribution of the polar Solomon rings, we performed both OOP and IP PFM measurements. To confirm the IP polarization projections of Solomon I and Solomon II polar textures, the test specimens were rotated by 0°, 45°, 90° or −90°, and 135° to the cantilever. During the measurement, a commercial Pt-Ir coated tip (Nanoword) was used. The AC voltage is 1.5 Vpp, and the frequency was ~22 kHz.

### TEM measurement

The sample for STEM was obtained by focused ion beam milling via the Precision Ion Polishing System (Model 691, Gatan Inc.). High-angle annular dark-field (HAADF) images were collected at 300 kV by using an aberration-corrected FEI Titan Themis G2 with spatial resolutions up to 60 pm.

### Macroscopic and microscopic IR absorption measurement

To have a first glimpse of the IR absorption for rhombohedral $BiFeO_3$, FTIR was carried out based on a Nicolet iS50 spectrometer (Thermo Fisher) under grazing incidence mode with IR beam polarized vertical to the substrate plane. Accordingly, we observed strong absorption at ~500 cm⁻¹, for the rhombohedral continuous thin film. To further distinguish the IR absorption for rhombohedral $BiFeO_3$ nanocrystals with upward, downward quad-domains and the Solomon topological polar structure, AFM-IR with an ultra-high spatial resolution is carried out as shown in Fig. 4 in the main text.

### Phase-field simulations

In our phase-field simulation, the polarization $P_i(P_x, P_y, P_z)$ can be described by evolving time-dependent Ginzburg-Landau equation[42,43] in a $BiFeO_3$ nanocrystal.

$$\frac{\partial P_i(r,t)}{\partial t} = -L \frac{\delta F_p}{\delta P_i(r,t)}, (i = x, y, z) \tag{3}$$

where $t$, $L$, and $F_p$ are simulation time, kinetic coefficient, and total free energy, respectively. Total free energy is defined as:

$$F_P = \iiint_V (f_{bulk}(\boldsymbol{P}_i) + f_{grad}(\boldsymbol{P}_{i,j}) + f_{elas}(\boldsymbol{P}_i, \varepsilon_{ij}) + f_{elec}(\boldsymbol{P}_i, \boldsymbol{E}_i))dV \tag{4}$$

where $f_{bulk}$, $f_{grad}$, $f_{elas}$, and $f_{elec}$ are bulk free-energy density, gradient energy density, elastic energy density, and electric energy density, respectively. The bulk free-energy density $f_{bulk}$ is a sixth-order polynomial,

$$f_{bulk} = a_1\left(P_x^2 + P_y^2 + P_z^2\right) + a_{11}\left(P_x^4 + P_y^4 + P_z^4\right)$$
$$+ a_{12}\left(P_x^2 P_y^2 + P_y^2 P_z^2 + P_z^2 P_x^2\right)$$
$$+ a_{111}\left(P_x^6 + P_y^6 + P_z^6\right) + a_{112}(P_x^4(P_y^2 + P_z^2) + P_y^4(P_z^2 + P_x^2) + P_z^4(P_x^2 + P_y^2))$$
$$+ a_{123}P_x^2 P_y^2 P_z^2 \tag{5}$$

where $\alpha_1, \alpha_{11}, \alpha_{12}, \alpha_{111}, \alpha_{112}$ and $\alpha_{123}$ are dielectric stiffness and higher-order stiffness under stress-free conditions. Among them, only $\alpha_1$ is

temperature-dependent, $\alpha_1 = (T - T_0)/(2\varepsilon_0 C_0)$, where $T$ is temperature, $T_0$ is Curie temperature, $\varepsilon_0$ (=$8.85 \times 10^{-12}$ F/m) is the dielectric permittivity of vacuum and $C_0$ is the Curie constant.

The gradient energy density is described in terms of polarization gradients. For simplicity, the gradient energy is taken to be isotropic, given as:

$$f_{grad} = \frac{1}{2}g_{ijkl}\boldsymbol{P}_{i,j} \tag{6}$$

where the $g_{ijkl}$ is gradient energy coefficient and $\boldsymbol{P}_{i,j} = \partial P_i/\partial x_j$[44]. In Eq. (6) and equations below, we adopt the Einstein summation convention in which repeated indices in a term imply summation. The elastic energy density can be written as,

$$f_{elas} = \frac{1}{2}c_{ijkl}e_{ij}e_{kl} = \frac{1}{2}c_{ijkl}\left(\varepsilon_{ij} - \varepsilon_{ij}^0\right)\left(\varepsilon_{kl} - \varepsilon_{kl}^0\right) \tag{7}$$

where the $c_{ijkl}$ is elastic stiffness tensor, $e_{ij}$ is elastic strain, $\varepsilon_{ij}$ is total elastic strain, $\varepsilon_{ij}^0$ is the stress-free strain given as $\varepsilon_{ij}^0 = Q_{ijkl}P_k P_l$, where $Q_{ijkl}$ represents the electrostrictive coefficient[45]. The electrostatic energy density, $f_{elec}$, is given by,

$$f_{elec} = -\boldsymbol{P}_i\boldsymbol{E}_i - \frac{\varepsilon_0\varepsilon_r}{2}\boldsymbol{E}_i\boldsymbol{E}_i \tag{8}$$

where $\varepsilon_r$ is background dielectric constants and $E_i$ is the total electric field. For the calculation of results presented in this manuscript, a $BiFeO_3$ nanocrystal with 45° edge-tilt was simulated. The simulation was performed with an eight-domain initial structure, strain-free film, and electrically charged edges, using the code we have developed to solve the above model. The value of coefficients involved in the present work is listed in Supplementary Table 1[46]. The simulation size is $93 \times 104 \times 20$ nm³, and the island boundary conditions are adopted. In the phase-field simulations, an electric potential is required to stabilize the center-type quad-domains, but no electric potential is required in the Solomon rings. The detailed island boundary conditions are shown in supplementary Fig. 6.

## Data availability

The data supporting the findings of this study are available within the article and its Supplementary Information.

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

## Acknowledgements

This work was supported by the Basic Science Centre Program of NSF of China (Grant No. 52388201), the NSF of China (Grant No. 12004036, 51972028, 51922055, 52225205), State Key Laboratory of New Ceramic and Fine Processing Tsinghua University (No. KFZD202201), and National Key Research and Development Program of China (Grant No. 2019YFA0307900). The work at Penn State is supported by the National Science Foundation through Grant No. DMR-2133373 and the Hamer Foundation through the Hamer Professorship. The authors also thank Dr. S. M. Chi from Beijing Institute of Technology and Dr. Y. J. Li from Bruker company for the discussions on AFM-IR measurement.

## Author contributions

H. H., and C.-W.N. conceived the project. J. W., Ji M., Y.F., H.Y. performed PFM measurements under the supervision of Jing M., and C.-W.N. Deshan L., H.M.J., S.D., C.G., and Di L. performed phase-field simulations under the supervision of H.H.. Ji M. fabricated the nanocrystals. Q.Z. performed TEM measurements under the supervision of L.G. Y.W. analyzed the TEM results under the supervision of Q.Z., Jing M., and L.G. J.W. and H.H. wrote the first draft of the manuscript. X.W., J.H., N.Z., D.Y., J.Z., Y.L., and L.C. discussed the results. All authors discussed the results and edited the manuscript.

## Competing interests

The authors declare no competing interests.
