## [Peer Review File · Nature Communications]

Polar Solomon Rings in Ferroelectric NanocrystalsEditorial Note: Parts of this Peer Review File have been redacted as indicated to remove third-party material where no permission to publish could be obtained.

REVIEWER COMMENTS

Reviewer #1 (Remarks to the Author):

Novel polar topologies in nano-scale oxide materials receive much attention nowadays. The authors here have made new progress. By successful preparations of the unique BFO nano-arrays, they have identified a new type of polar topology here, that is, the polar Solomon rings, which are composed of two interwoven ring-like polarizations in the special R BFO nano-island. Moreover, by applying external electric fields, reversible transformations between these polar structures and domain vertex textures have further been proved via PFM measurements. They also found that the Solomon rings and quad-domains show different absorptions of terahertz infrared waves and infrared display behaviors. It is somewhat unbelievable since in a general thinking, it will tend to obtain some center-typed domains in these BFO islands, which are common in previous works. Thus, this result is stimulating for other ferroelectric nanocrystals. I believe this observation will be illuminative for future studies on the searching and applications of novel polar topologies. Despite of the interesting results, I also have some concerns..

1.I noticed that the obtained BFO nano-arrays were grown by PLD, which are important for the formation of the polar Solomon rings. But how to get this kind of BFO nano-arrays is still not clear. Is the LAO substrate important here? Are there some special growth rates used here? (Since the different growth rates may induce completely different domain structures in BFO on LAO as reported previously) From Fig. S4, it seems that on the LAO substrate, some areas are absent of BFO (left in Fig. S4a)? Moreover, is the BFO/LSMO/LAO interface relaxed? The author claimed that "To stimulate island-growth mode for BiFeO₃ nanocrystals, ~2 nm thick (La,Sr)MnO₃ buffer layer was pre-grown on LaAlO₃ substrate to decrease the surface energy." It is well known that the (001) substrates usually favor to the growth of high-quality perovskite films due to the relatively lower surface energy of the perovskite (001) facet. If the introduction of LSMO would reduce the surface energy, layer-growth mode will be assisted rather than island-mode. More discussion is needed to interpret this issue.

2.The authors present high-quality HAADF images shown in Fig.S4, which is useful and important for studying the domain structures. Is there any evidence from these HAADF-STEM imagings to reveal the strain distributions like in Fig. S9? The combinations of HAADF-STEM and GPA will be effective for extracting these strains experimentally, which will be useful at the scales studied here. While the PFM may be inefficient for extracting domain structures in a 3D fashion, the experimental strain maps from STEM may help make the results more convincing. In addition to the cross-sectional HAADF-STEM images in Fig.S4, I suggest that atomically resolved HAADF-STEM imaging from plan-view observations should also be provided to see directly the configurations of Solomon rings. At least the intersection of interwoven rings could possibly be imaged. Recently, the 4D-STEM technique is gradually acting as the powerful tool. Whether the Solomon ring could be directly determined by the 4D-STEM technique, which would be also largely helpful.

3.The results of in-plane PFM phase images in Fig. S3, Fig. S8, Fig. S10, Fig. S11 and Fig. 3 are not clear enough to identify the details. The determination of in-plane polarization direction is not self-consistent to some extent in these figures. Please provide raw data of these figures in the supplementary files. By the way, how the authors relocate the same nanocrystal after rotating by different angles?

4.In Fig. 5g-i, the authors mentioned that the 'BIT' symbol of the quad-domain state was poled in the Solomon matrix. The IP phase image after poling should also be provided. In

addition, the result of IR absorption image in Fig. 5i is a little bit confusing. The IR absorption intensity seems to be influenced by the morphology of the BFO films. As shown in Fig. 5i, the regions between nanocrystals display the higher absorption intensity (white) than that of unpoled nanocrystals. Please give some discussions.

5. I notice that some of the authors also participated in a previous work about the center-type domains found in the similar systems (BiFeO₃ nanodots grown on LaAlO₃ substrate) [Ma, J. et al. Nat. Nanotech. 13, 947 (2018)]. What is the difference in the growth preparation/simulation conditions of the two works? What is the ground state of a BiFeO₃ nanodot? The authors mentioned they adopted the island boundary conditions. What are the island boundary conditions?

6. In addition to the PFM measurements, the Solomon ring structure is mainly demonstrated by shrinking the simulated domain loops in the present manuscript. The authors artificially create an initial eight-domain structure in the phase-field model. Does it mean that the authors created a Solomon ring in the nano crystal?

7. How to judge the display property of a material/device? The authors could compare the reported topological structure with other display objects in the discussion.

8. From Fig. 2b, it seems that the center vortex in L4 is somewhat large, with the size of about 20 nm, which means that Solomon rings may be absent here. How to explain this? Why the region of $W=1$ is so small in the winding number mapping of Fig. 2g?

Reviewer #2 (Remarks to the Author):

The authors report a topological structure named polar Solomon rings and its reversible phase transition triggered by the electric field in BiFeO₃ ferroelectric nanocrystal. The two types of topological polar textures exhibit distinct absorption of terahertz infrared waves, which shows potential applications in infrared displays with a nanoscale resolution. Their study confirms the existence and electric manipulation of polar Solomon rings, laying the foundations for its promising future. However, the following comments need to be addressed prior to publication:

Comment 1: The authors carefully characterize the Solomon rings topological polar structures in BiFeO₃ thin films. However, as part of the comprehensive understanding, the origin of this phenomenon should be discussed. For example, as the nano-scale thin film, the role of the depolarization field for forming the polar structure should be addressed. Meanwhile, the strain confinement of the substrate and the dipole interaction can also play important characters.

Comment 2: The authors should provide more information or references to clarify the interaction between the anisotropic adsorption of the terahertz infrared waves and the two distinct ferroelectric polar structures. The correlation between the intensity of absorption and ferroelectric orientation and polar gradient (domain wall) should be discussed further in the article. Meanwhile, it will be more convincing if the details about the polar orientation of the two structures are provided. A more considerable explanation of the absorption difference between the two polar structures could be drawn with such.

Comment 3: Owing to the potential application in the high-resolution display area caused by the particular polar topological structure contrast, the relaxation time of the topological phase transition could be a key point for the display speed and performance. The authors are encouraged to provide some information about this issue.

Comment 4: The electric boundary condition and the depolarization field (bounded charge) screening condition of all experiments and simulations should be provided in the article for better reproductions of the results by others.

Comment 5: It is better to see a consistent and sequent arrangement of the subfigures in all figures.

Reviewer #3 (Remarks to the Author):

In this work, the authors present an interesting study on the observation of polar Solomon rings in a ferroelectric nanocrystal. Polar Solomon rings are demonstrated, by combining piezoresponse force microscopy observations and phase-field simulations, to be reversible switching to vertex textures by an electric field. The two types of topological polar textures exhibit distinct absorption of terahertz infrared waves, which can be utilized in infrared displays with a nanoscale resolution. I think this work should be published. There are, however, a few concerns and I'd like to see them be addressed before the final consideration of publication.

1. In this study, the authors used toroidal moment and winding number to confirm the topological characteristic of the polar Solomon. Specifically, the Solomon state possesses a non-zero value of G_z and a winding number of 1 in the center of Solomon rings. However, I was not convinced about the topological characterization of the Solomon state, since the common polarization vortex also has the same characteristics, i.e., a non-zero value of G_z and a winding number of 1. The author should introduce a suitable parameter to characterize the polar Solomon.
2. The author should clarify and discuss the forming mechanism of the polar Solomon. Such a mechanism is critical to reproduce and controlling the polar Solomon formation. In addition, the roles of geometry, ferroelectric phase, epitaxial strain, or flexoelectric effect in the formation of polar Solomon should be discussed.
3. The authors demonstrated the reversible switching between polar Solomon rings and vertex textures by an electric field. However, the rotation of the original and re-formed Solomon rings are the same. Is there any possible approach to switch the orientation of Solomon rings, specifically, from clockwise to counter-clockwise and vice versa?
4. It is seemingly that the polar Solomon rings can only be formed in rhombohedral phase ferroelectric. Is it possible to extend the idea of polar Solomon to other ferroelectric phases? The authors should discuss conditions to stabilize the polar Solomon.
5. Regarding the phase field simulation, the authors stated that "the simulation was performed with an eight-domain initial structure, ...". I guess the eight-domain initial structure is close to the polar Solomon state, such that it is easy to get the polar Solomon as a stable state. However, is it possible to stabilize the Solomon state in the phase field simulation by starting from a random distribution of the polarization field?

A point-by-point response letter (for NCOMMS-22-52845)

We greatly appreciate the referees' positive recommendations, valuable comments, and suggestions. Accordingly, we have carried out extended experiments and simulations, and made appropriate revisions as discussed below, which have greatly improved our manuscript. Below are our point-by-point responses to the referees' comments and the changes (in blue) made in the revised manuscript.

Response to the referee #1

Reviewer #1 (Remarks to the Author):

Novel polar topologies in nano-scale oxide materials receive much attention nowadays. **The authors here have made new progress.** By successful preparations of the unique BFO nano-arrays, they have identified a new type of polar topology here, that is, the polar Solomon rings, which are composed of two interwoven ring-like polarizations in the special R BFO nano-island. Moreover, by applying external electric fields, reversible transformations between these polar structures and domain vertex textures have further been proved via PFM measurements. They also found that the Solomon rings and quad-domains show different absorptions of terahertz infrared waves and infrared display behaviors. It is somewhat unbelievable since in a general thinking, it will tend to obtain some center-typed domains in these BFO islands, which are common in previous works. Thus, **this result is stimulating for other ferroelectric nanocrystals.** I believe **this observation will be illuminative for future studies on the searching and applications of novel polar topologies.** Despite of the interesting results, I also have some concerns.

Response:

We are very grateful to the referee for highlighting the major discoveries in our work, and for his/her positive viewpoint that “the authors here have made new progress”, “this result is stimulating for other ferroelectric nanocrystals” and “this observation will be illuminating for future studies on the search and applications of novel polar topologies”. We have addressed all the concerns as discussed below.

1. I noticed that the obtained BFO nano-arrays were grown by PLD, which are important for the formation of the polar Solomon rings. But how to get this kind of BFO nano-arrays is still not clear. Is the LAO substrate important here? Are there some special growth rates used here? (Since the different growth rates may induce completely different domain structures in BFO on LAO as reported previously) From Fig. S4, it seems that on the LAO substrate, some areas are absent of BFO (left in Fig. S4a)? Moreover, is the BFO/LSMO/LAO interface relaxed? The author claimed that “To stimulate island-growth mode for BiFeO₃ nanocrystals, ~2 nm thick (La,Sr)MnO₃ buffer layer was pre-grown on LaAlO₃ substrate to decrease the surface energy.” It is

well known that the (001) substrates usually favor to the growth of high-quality perovskite films due to the relatively lower surface energy of the perovskite (001) facet. If the introduction of LSMO would reduce the surface energy, layer-growth mode will be assisted rather than island-mode. More discussion is needed to interpret this issue.

Response:

We thank the referee for bringing up this great point about how to get this nano-island array.

Our experiments showed that the LSMO wetting layer and the growth temperature are important for the nano-island growth mode.

First, our previous first-principle calculations showed that the surface energy of the LSMO changes with its thickness [1]. When the thickness of LSMO is ~2-12 nm, the surface energy of BFO ($\sim 1.0 \text{ J m}^{-2}$) is larger than that ($\sim 0.47 \text{ J m}^{-2}$) of the (LSMO)/LAO system, which prefers the island growth mode as observed (see Fig. R1(b-e)). However, when the thickness of LSMO is smaller than 2 nm or larger than 12 nm, the surface energy of BFO ($\sim 1.0 \text{ J m}^{-2}$) is smaller than that ($1.3\text{-}1.4 \text{ J m}^{-2}$) of the LSMO/LAO system, which prefers the layer growth mode as observed (see Fig. R1(a, f)) [1].

[Redacted]

Fig. R1. Evolution of BFO morphology in BFO/LSMO/LAO heterostructures with increasing LSMO thickness observed previously [1]. BFO nano-islands are formed when the LSMO thickness is between 2 nm and 12 nm.

Second, the high growth temperature ($\sim 680\text{-}720 \text{ }^\circ\text{C}$) helps the formation of periodically arranged dislocations at the BFO/LSMO interface (Fig. R2), which on the one hand relax the epitaxial strain and on the other hand induce the nucleation of the rhombohedral-phase nano-island array (Fig. R3).

[Redacted]

Fig. R2. Strain and dislocation characterization in a self-assembled BFO nanocrystal [2]. **a**, Cross-sectional STEM image of a BFO nanocrystal. **b**, Enlarged image near the BFO/LSMO interface, where interface dislocations are marked by white arrows. **c**, Geometric phase analysis (GPA) for the BFO nanocrystal along $[100]_{pc}$ direction, where the dislocations are marked by blue and yellow arrows. **d**, GPA for the enlarged image in **(b)**. The inset in **(b)** is the enlarged atomic resolution image of the rhombohedral-phase BFO, where the light and dark spheres represent Bi and Fe, respectively.

[Redacted]

Fig. R3. Evolution of BFO morphology in BFO/LSMO/LAO heterostructures with increasing growth temperature [1]. The BFO nano-islands appear between 680 °C and 720 °C. At 600 °C, BFO shows a layer growth mode. At 750 °C, needle-like nanostructures appear due to the evaporation of bismuth elements.

The above discussion on how to get this kind of BFO nano-island arrays were reported in our previous work [1, 2]. For clear illustration, we have added some discussion in the revised manuscript (see the Methods section).

References:

- [1] Self-assembly growth of a multiferroic topological nanoisland array, *Nanoscale* **11**, 20514 (2019).
- [2] Stabilization of ferroelastic charged domain walls in self-assembled BiFeO_3

2. The authors present high-quality HAADF images shown in Fig.S4, which is useful and important for studying the domain structures. Is there any evidence from these HAADF-STEM imagings to reveal the strain distributions like in Fig. S9? The combinations of HAADF-STEM and GPA will be effective for extracting these strains experimentally, which will be useful at the scales studied here. While the PFM may be inefficient for extracting domain structures in a 3D fashion, the experimental strain maps from STEM may help make the results more convincing. In addition to the cross-sectional HAADF-STEM images in Fig.S4, I suggest that atomically resolved HAADF-STEM imaging from plan-view observations should also be provided to see directly the configurations of Solomon rings. At least the intersection of interwoven rings could possibly be imaged. Recently, the 4D-STEM technique is gradually acting as the powerful tool. Whether the Solomon ring could be directly determined by the 4D-STEM technique, which would be also largely helpful.

Response:

We thank the referee for the good suggestions.

One thing we should note is that polar Solomon is a three-dimensional structure, which cannot be fully revealed by the strain mapping in any two-dimensional plane. Therefore, we performed GPA on the cross-sectional HAADF image, and the result is consistent with the phase-field simulations as shown in Fig. R4, which has been added in Fig. S5.

Fig. R4. Cross-sectional strain distribution of a BFO nanocrystal. **a**, GPA image for a cross-sectional BFO nanocrystal. **b**, Phase-field simulations for strain distribution in a BFO nanocrystal.

3. The results of in-plane PFM phase images in Fig. S3, Fig. S8, Fig. S10, Fig. S11 and Fig. 3 are not clear enough to identify the details. The determination of in-plane polarization direction is not self-consistent to some extent in these figures. Please provide raw data of these figures in the supplementary files. By the way, how the authors relocate the same nanocrystal after rotating by different angles?

Response:

Thank the reviewer for the comment. We have provided the corresponding raw data in the supplementary files (see supplementary Fig. S3, Fig. S11, Fig. S14, Fig. S15, Fig. S16), which can also be seen in Figs. R5-R9 below.

In addition, we combined optical microscopy and atomic force microscopy (AFM) to relocate the same nanocrystal. Before rotating the sample, we first snapshot a 20 μm

AFM image and a 100 μm optical microscope image to mark the interested nanocrystal.

Fig. R5. PFM raw data of the OOP and IP polarization projection for polar Solomon rings with LK of +2. **a, b**, Morphology and IP PFM phase image for the BiFeO_3 nanocrystal in the initial state. **c-h**, OOP (**c, e, g**) and IP (**d, f, h**) PFM phase images for the BiFeO_3 nanocrystal when the sample is rotated by -90° (**c, d**), 45° (**e, f**) and 135° (**g, h**) with respect to the cantilever. This BiFeO_3 nanocrystal is the same as that shown in Fig. 1d in the main text.

Fig. R6. PFM raw data of the OOP and IP polarization projection for polar Solomon rings with LK of -2. **a, b**, Morphology and IP PFM phase image for the BiFeO_3 nanocrystal in the initial state. **c-h**, OOP (**c, e, g**) and IP (**d, f, h**) PFM phase images for the BiFeO_3 nanocrystal when the sample is rotated by 90° (**c, d**), 45° (**e, f**) and 135° (**g, h**) with respect to the cantilever.

Fig. R7. PFM raw data of the OOP and IP polarization projection for polar Solomon rings with LK of +2. **a, b**, OOP (**a**) and IP (**b**) PFM phase images for the BiFeO₃ nanocrystal in the initial state. **c, d**, OOP (**c**) and IP (**d**) PFM phase images for the BiFeO₃ nanocrystal when the specimen is rotated by 45° with respect to the cantilever. **e, f**, OOP (**e**) and IP (**f**) PFM phase images for the BiFeO₃ nanocrystal when the specimen is rotated by 90° with respect to the cantilever. **g, h**, OOP (**g**) and IP (**h**) PFM phase images for the BiFeO₃ nanocrystal when the specimen is rotated by 135° with respect to the cantilever.

Fig. R8. PFM raw data of the OOP and IP polarization projection for polar Solomon rings with LK of -2. **a, b**, OOP (**a**) and IP (**b**) PFM phase images for the BiFeO₃

nanocrystal in the initial state. **c, d**, OOP (**c**) and IP (**d**) PFM phase images for the BiFeO₃ nanocrystal when the specimen is rotated by 45° with respect to the cantilever. **e, f**, OOP (**e**) and IP (**f**) PFM phase images for the BiFeO₃ nanocrystal when the specimen is rotated by 90° with respect to the cantilever. **g, h**, OOP (**g**) and IP (**h**) PFM phase images for the BiFeO₃ nanocrystal when the specimen is rotated by 135° with respect to the cantilever.

Fig. R9. Electric-field driven topological phase transition between polar Solomon rings and vertex structures in a BiFeO₃ nanocrystal. **a, b**, OOP (**a**) and IP (**b**) PFM phase images for the domain pattern in a BiFeO₃ nanocrystal when the electric fields of 0 V, -4 V, 2 V, 3 V, and -2 V are applied to the scanning probe, respectively.

4. In Fig. 5g-i, the authors mentioned that the ‘BIT’ symbol of the quad-domain state was poled in the Solomon matrix. The IP phase image after poling should also be provided. In addition, the result of IR absorption image in Fig. 5i is a little bit confusing. The IR absorption intensity seems to be influenced by the morphology of the BFO films. As shown in Fig. 5i, the regions between nanocrystals display the higher absorption intensity (white) than that of unpoled nanocrystals. Please give some discussions.

Response:

We thank the referee for the good suggestion. We have provided the IP phase image after poling in Fig. S22, which can also be seen in Fig. R10 below.

Fig. R10. In-plane PFM phase image for BiFeO₃ nanocrystals after poling. **a, b**, A large scale (**a**) and magnified (**b**) IP PFM phase image for the upward quad-domains when the sample is positioned at 45° with respect to the cantilever. **c, d**, A large scale (**c**) and

magnified (d) IP PFM phase image for the upward quad-domains when the sample is positioned at 135° with respect to the cantilever.

As regards Fig. 5i, we have added additional experimental result and discussion of the AFM-IR absorption image. In fact, the rhombohedral BFO (R-BFO) nano-islands are embedded in the tetragonal BFO (T-BFO) continuous thin films. According to the FTIR spectrum, the T-BFO thin film shows higher absorption intensity than the R-BFO thin film around 550 cm^{-1} as shown in Fig. R11, which explains the higher absorption intensity (white) in the regions (e.g., T-BFO thin film) between nanocrystals than that of unpoled nanocrystals. The above discussion has been added in the revised manuscript (page 6, paragraph 1) and Fig. S18.

Fig. R11. FTIR absorption spectrum for T-BFO and R-BFO thin film.

5. I notice that some of the authors also participated in a previous work about the center-type domains found in the similar systems (BiFeO₃ nanodots grown on LaAlO₃ substrate) [Ma, J. et al. Nat. Nanotech. 13, 947 (2018)]. What is the difference in the growth preparation/simulation conditions of the two works? What is the ground state of a BiFeO₃ nanodot? The authors mentioned they adopted the island boundary conditions. What are the island boundary conditions?

Response:

We thank the referee for the good comment.

In the experiments, the growth preparation conditions of the two works are nearly the same. The ground state of the a BiFeO₃ nanodot is the Solomon rings, the center-type domains are obtained by applying an out-of-plane electric field.

In the phase-field simulations, the detailed dimensions of the nano-island are shown in Fig. R12(a) below, and the nano-island is surrounded by air. For the ground state Solomon rings, no electric potential is required for the nano-island, see Fig. R12(b). However, for the center-type quad-domains, an electric potential is required to stabilize it, see Fig. R12(c). These have been added in Fig. S6 and discussed in Methods section.

Fig. R12. Island boundary conditions used in phase-field simulations. **a**, The dimensions of the island. The island boundary conditions for **b**, Solomon rings and **c**, quad-domains.

6. In addition to the PFM measurements, the Solomon ring structure is mainly demonstrated by shrinking the simulated domain loops in the present manuscript. The authors artificially create an initial eight-domain structure in the phase-field model. Does it mean that the authors created a Solomon ring in the nanocrystal?

Response:

We thank the referee for the good comment. The Solomon rings can also develop from a random domain. As shown in Fig. R13 below, a random domain is transformed into quad-domains after an out-of-plane electric field of 8 MV/m is applied, which is further transformed into Solomon rings after a reversed electric field of -0.5 MV/m is applied to the nanocrystal.

Fig. R13. Domain evolution from a random state to Solomon states. After 1000 steps, an out-of-plane electric field of 8 MV/m is applied to the nanocrystal. After 4000 steps, a reversed electric field of -0.5 MV/m is applied to the nanocrystal.

7. How to judge the display property of a material/device? The authors could compare the reported topological structure with other display objects in the discussion.

Response:

We thank the referee for the good suggestion.

The size of the nanocrystals is 200-300 nm, and each nanocrystal can serve as an addressable pixel, which is much smaller than the pixel size (3 μm -200 μm) of the other display objects [3-7]. The above discussion has been added in the revised manuscript

(page 6, paragraph 1).

References:

[3] Covert infrared image encoding through imprinted plasmonic cavities, *Light: Science & Applications* **7**, 93 (2018). (pixel size: $\sim 10 \mu\text{m}$)

[4] Vertical full-colour micro-LEDs via 2D materials-based layer transfer, *Nature* **614**, 81 (2023). (pixel size: $\sim 14 \mu\text{m}$)

[5] A multi-directional backlight for a wide-angle, glasses-free three-dimensional display, *Nature* **495**, 348 (2013). (pixel size: $\sim 200 \mu\text{m}$)

[6] Metasurface-driven OLED displays beyond 10,000 pixels per inch, *Science* **370**, 459 (2020). (pixel size: $\sim 3 \mu\text{m}$)

[7] Imaging-based molecular barcoding with pixelated dielectric metasurfaces, *Science* **360**, 1105 (2018). (pixel size: $\sim 100 \mu\text{m}$)

8. From Fig. 2b, it seems that the center vortex in L4 is somewhat large, with the size of about 20 nm, which means that Solomon rings may be absent here. How to explain this? Why the region of $W=1$ is so small in the winding number mapping of Fig. 2g?

Response:

Thank the referee for the comment. The polar Solomon rings are a 3D topological structure, they cannot disappear in a plane. In this 3D topological structure, R_1^- , R_2^- , R_3^- and R_4^- are positioned in the center from the top surface, while R_1^+ , R_2^+ , R_3^+ and R_4^+ are positioned in the center from the bottom surface, as shown in Fig. R14 below. These eight domains are twisted in the 3D space and form Solomon rings, see Fig. R15 below. Fig. R15 has been added in Fig. S7.

Fig. R14. **a**, BiFeO₃ nano-island with eight domain variants. **b**, **d**, Four domain variants with downward polarization direction. **c**, **e**, Four domain variants with upward polarization direction.

Fig. R15. **a**, BiFeO₃ nano-island with Solomon rings. **b, c**, The extracted red and blue rings from (a). **d**, Schematic illustration of the interwoven way of the blue and red rings in 3D space for a BiFeO₃ nanocrystal. The white arrows indicate the polarization direction of the domains.

To calculate the winding number, we use the following equation,

$$n = \frac{1}{2\pi} \int_0^{2\pi} \frac{d\phi}{d\theta} (r, \theta) d\theta \quad (1)$$

and the region of $W=1$ requires eight domains to coexist. Thus, it shows a very small region, which is consistent with the results in other reports [8].

References:

[8] Configurable topological textures in strain graded ferroelectric nanoplates. *Nature Communications* **9**, 403 (2018).

Response to the referee #2

Reviewer #2 (Remarks to the Author):

The authors report a topological structure named polar Solomon rings and its reversible phase transition triggered by the electric field in BiFeO₃ ferroelectric nanocrystal. The two types of topological polar textures exhibit distinct absorption of terahertz infrared waves, which shows potential applications in infrared displays with a nanoscale resolution. **Their study confirms the existence and electric manipulation of polar Solomon rings, laying the foundations for its promising future.** However, the following comments need to be addressed prior to publication:

Response:

We really appreciate the referee's positive summary.

Comment 1: The authors carefully characterize the Solomon rings topological polar structures in BiFeO₃ thin films. However, as part of the comprehensive understanding, the origin of this phenomenon should be discussed. **For example, as the nano-scale thin film, the role of the depolarization field for forming the polar structure should be addressed.** Meanwhile, the **strain confinement of the substrate and the dipole interaction** can also play important characters.

Response:

We thank the referee for pointing this out. We have performed additional phase-field simulations without considering the depolarization field as shown in Fig. R16 below, which shows that the Solomon rings did not form, indicating that the depolarization field is important for the formation of this polar structure. On the other hand, the strain is completely relaxed at the island/substrate interface, which is demonstrated by the GPA image for the HAADF image in our previous work [2], also shown in Fig. R17 below. The above discussion has been added in the revised manuscript (page 3, paragraph 3), and Fig. R16 has been added in Fig. S13.

Fig. R16. **a**, Phase-field simulation of the domain pattern in a BFO nanocrystal without consideration of the depolarization field. **b**, Phase-field simulation of the domain pattern in a BFO nanocrystal with consideration of the depolarization field.

[Redacted]

Fig. R17. Strain and dislocation characterization in a self-assembled BFO nanocrystal [2]. **a**, Cross-sectional STEM image of a BFO nanocrystal. **b**, Enlarged image near the BFO/LSMO interface, where interface dislocations are marked by white arrows. **c**,

GPA analysis for the BFO nanocrystal along $[100]_{pc}$ direction, where the dislocations are marked by blue and yellow arrows. **d**, GPA analysis for the enlarged image in (b). The inset in (c) is the enlarged atomic resolution image of the R-phase BFO, where the light and dark sphere represent Bi and Fe, respectively.

References:

[2] Stabilization of ferroelastic charged domain walls in self-assembled BiFeO_3 nanoislands, *Journal of Applied Physics* **128**, 124103 (2020).

Comment 2: The authors should provide more information or references to clarify the interaction between the anisotropic adsorption of the terahertz infrared waves and the two distinct ferroelectric polar structures. The correlation between the intensity of absorption and ferroelectric orientation and polar gradient (domain wall) should be discussed further in the article. Meanwhile, it will be more convincing if the details about the polar orientation of the two structures are provided. A more considerable explanation of the absorption difference between the two polar structures could be drawn with such.

Response:

We thank the referee for pointing this out. The absorption of the THz electromagnetic wave is related to the relative angle between the ferroelectric polarization and the electromagnetic wave polarization. If they are parallel or antiparallel, the absorption is strongest, but if they are perpendicular to each other, the absorption is weakest [9].

In addition, domain walls are inactive in the THz band due to their long relaxation time, and they act as defective interfaces arising from their frustrated dipoles. These unfulfilled dipoles in the domain walls contribute less to the permittivity than their counterparts in the domains [10], as shown in Fig. R18 below. Therefore, if the volume fraction of the domain walls is large, the absorption intensity of the THz electromagnetic wave will decrease.

[Redacted]

Fig. R18. Illustration of different permittivity in ferroelectric crystals with distinct volume fraction of domain walls [10]. **a**, Schematic setup of the THz reading technique. **b**, Domain wall density change in depoled and poled ferroelectric crystals. **c**, THz

reading of different permittivity in depoled and poled ferroelectric crystals.

The electromagnetic wave we used is out-of-plane polarization, see Fig. R19a. For the two polar structures, e.g., Solomon rings (Fig. R19b) and quad-domains (Fig. R19c,d), their out-of-plane polarization projection is either parallel or antiparallel to the electromagnetic polarization, but the main difference is the volume fraction of the domain walls. For the quad-domains, there are 4 domain variants and the volume fraction of the domain walls is small. However, for Solomon rings, the eight domain variants are twisted in the 3D space, inducing a larger volume fraction of the domain walls than the quad-domains. Thus, we observe a weaker THz wave absorption in Solomon rings than in the quad domains.

Fig. R19. **a**, Schematic of the experimental setup with OOP electric field vector. **b**, Solomon rings with a large volume fraction of the domain walls. **c**, **d**, Quad-domains with a small volume fraction of the domain walls.

The above discussion has been added in the revised manuscript (page 5, paragraph 2). Figure R19 has been added in Fig. S19.

References:

- [9]. M. Guo et al., Toroidal polar topology in strained ferroelectric polymer. *Science* **371**, 1050-1056 (2021).
- [10] M. Zhang et al., Terahertz Reading of Ferroelectric Domain Wall Dielectric Switching. *ACS Appl Mater Interfaces* **13**, 12622-12628 (2021).

Comment 3: Owing to the potential application in the high-resolution display area caused by the particular polar topological structure contrast, the relaxation time of the topological phase transition could be a key point for the display speed and performance. The authors are encouraged to provide some information about this issue.

Response:

We thank the referee for the good suggestion. The relaxation time of the topological phase transition depends on the ferroelectric polarization switching time, which has a time constant of sub-nanoseconds [11-12], indicating the ultrafast conversion speed for the displayed images. The above discussion has been added in the revised manuscript (page 6, paragraph 1).

References:

[11] Ultrafast Polarization Switching in Thin-Film Ferroelectrics, *Applied Physics Letters*, **84**, 1174-1176 (2004).

[12] Sub-Nanosecond Memristor Based on Ferroelectric Tunnel Junction, *Nat Commun*, **11**, 1439 (2020).

Comment 4: The electric boundary condition and the depolarization field (bounded charge) screening condition of all experiments and simulations should be provided in the article for better reproductions of the results by others.

Response:

We thank the referee for the good suggestion. In the experiment, we used 2nm LSMO as the bottom electrode.

In the phase-field simulations, an electric potential is required to stabilize the center-type quad-domains, but no electric potential is required in the Solomon rings. The detailed island boundary conditions are shown in Fig. R20 below. This information has been added in Fig. S6 and discussed in the Methods section.

Fig. R20. The island boundary conditions for **a** quad-domains and **b** Solomon rings.

Comment 5: It is better to see a consistent and sequent arrangement of the subfigures in all figures.

Response:

We thank the referee for the good suggestion, and we have changed the arrangement of the subfigures in the revised manuscript.

Response to the referee #3

Reviewer #3 (Remarks to the Author):

In this work, the authors present **an interesting study** on the observation of polar Solomon rings in a ferroelectric nanocrystal. Polar Solomon rings are demonstrated, by combining piezoresponse force microscopy observations and phase-field simulations, to be reversible switching to vertex textures by an electric field. The two types of topological polar textures exhibit distinct absorption of terahertz infrared waves, which can be utilized in infrared displays with a nanoscale resolution. **I think this work should be published.** There are, however, a few concerns and I'd like to see them be addressed before the final consideration of publication.

Response:

We greatly appreciate the referee's recommendation that our work be published. We have addressed the issues and concerns as discussed below.

1. In this study, the authors used toroidal moment and winding number to confirm the topological characteristic of the polar Solomon. Specifically, the Solomon state possesses a non-zero value of G_z and a winding number of 1 in the center of Solomon rings. However, I was not convinced about the topological characterization of the Solomon state, since the common polarization vortex also has the same characteristics, i.e., a non-zero value of G_z and a winding number of 1. The author should introduce a suitable parameter to characterize the polar Solomon.

Response:

Thank the referee for the good comment. We fully agree with the referee that a non-zero value of G_z can only indicate that there is an in-plane vortex, and a winding number of 1 can only show that the polar structure has a topological invariant.

We have discussed the topological properties of the polar Solomon rings based on the knots and links theory, a mathematical topological theory, in the Introduction section and Figs. 1, S1, and S2 (also in Fig. R21 and R22 below), which indicates the polar Solomon rings can be described as a 4_1^2 link.

Fig. R21. Stereoscopic representations of multiple knots and links. **a**, Unknot, 0_1 . **b**, Trefoil knot, 3_1 . **c**, Figure-eight knot, 4_1 . **d**, 5_1 . **e**, 5_2 . **f**, Two-component trivial link, 0_1^2 . **g**, Hopf link, 2_1^2 . **h**, Solomon link, 4_1^2 . **i**, Whitehead link, 5_1^2 . **j**, Star of David link, 6_1^2 .

Fig. R22. **a**, **b**, A 3D polar vortex composed of $R_4^-[\bar{1}\bar{1}\bar{1}]$, $R_3^+[\bar{1}\bar{1}\bar{1}]$, $R_2^-[\bar{1}\bar{1}\bar{1}]$, $R_1^+[111]$, and $R_4^+[\bar{1}\bar{1}\bar{1}]$, $R_3^-[\bar{1}\bar{1}\bar{1}]$, $R_2^+[\bar{1}\bar{1}\bar{1}]$, $R_1^-[\bar{1}\bar{1}\bar{1}]$ polarization variants in rhombohedral phase BiFeO₃. **c**, The definition of polar Solomon rings with the linking number (LN) of +2, from the mathematical point of view. **d**, Distorted 3D polar Solomon rings which can be considered as two intersecting vortices composed of eight polarization variants of BiFeO₃ as shown in (a) and (b). **e**, The construction of OOP and IP polarization projection for a BiFeO₃ nanocrystal by PFM measurement. The profile of the downward domain pattern is outlined by the white line to guide the eye. **f**, The extracted two intersecting polar vortices projected in the (001) plane.

2. The author should clarify and discuss the forming mechanism of the polar Solomon. Such a mechanism is critical to reproduce and controlling the polar Solomon formation. In addition, the roles of **geometry**, **ferroelectric phase**, **epitaxial strain**, or **flexoelectric effect** in the formation of polar Solomon should be discussed.

Response:

We thank the referee point out this important point.

The observed polar Solomon state is the ground state for the self-assembled BiFeO₃ nano-islands with the size of ~300 nm.

The epitaxial strain is relaxed by the arrayed dislocations at the film/substrate interface [2], as shown in Fig. R23 below. Normally, the flexoelectric effect takes effect

at several nanometer scales [13], so it takes negligible effect on the formation of Solomon rings due to the large size of the nano-island.

Fig. R23. Strain and dislocation characterization in a self-assembled BFO nanocrystal. **a**, Cross-sectional STEM image of a BFO nanocrystal. **b**, Enlarged image near the BFO/LSMO interface, where interface dislocations are marked by white arrows. **c**, GPA analysis for the BFO nanocrystal along $[100]_{pc}$ direction, where the dislocations are marked by blue and yellow arrows. **d**, GPA analysis for the enlarged image in **(b)**. The inset in **(c)** is the enlarged atomic resolution image of the R-phase BFO, where the light and dark sphere represent Bi and Fe, respectively.

Ferroelectric phase and geometry are two significant effects, as discussed below:

Rhombohedral ferroelectric phase owns eight polarization variants, they can form two Stereoscopic vortices in three-dimensional space. These two 3D vortices are two element rings for the formation of Solomon link, which is the same as the mathematic definition of a 4_1^2 link as shown in Fig. R22 above. The rectangular shape of the nano-island constrained the eight domain and make the two polar rings twisted and connected as a way of Solomon link.

We have added the above discussion in the revised manuscript (page 3, paragraph 3).

References:

[2] Stabilization of ferroelastic charged domain walls in self-assembled BiFeO_3 nanoislands, *Journal of Applied Physics* **128**, 124103 (2020).

[13] Wu, M., Zhang, X., Li, X. *et al.* Engineering of atomic-scale flexoelectricity at grain boundaries. *Nat. Commun.* **13**, 216 (2022).

3. The authors demonstrated the reversible switching between polar Solomon rings and vertex textures by an electric field. However, the rotation of the original and re-formed Solomon rings are the same. Is there any possible approach to switch the orientation of Solomon rings, specifically, from clockwise to counter-clockwise and vice versa?

Response:

We thank the referee for pointing out this important issue. We observed the

reversible switching between polar Solomon rings and vertex textures by an electric field. However, unfortunately, we do not observe switching of the orientation of Solomon rings by an electric field. Our phase-field simulations in the previous report [14] showed it is hard to switch the orientation of Solomon rings by an electric field. As we can see in Fig. R24 below, by using the surface charge and the electric field, the polarity of the vortex changes, but the vorticity does not change.

Fig. R24. **a** Schematic of surface charge application. **b** Schematic of electric field application. **c** Reversible vortex chirality evolution under the stimuli presented in **(a)**. **d** Reversible vortex chirality evolution under the stimuli presented in **(b)**. The * is a normalized value, $E^* = E \times 0.192$ MV/cm.

References:

[14] Phase-field simulations of vortex chirality manipulation in ferroelectric thin films. *npj Quantum Mater.* **7**, 34 (2022).

4. It is seemingly that the polar Solomon rings can only be formed in rhombohedral phase ferroelectric. Is it possible to extend the idea of polar Solomon to other ferroelectric phases? The authors should discuss conditions to stabilize the polar Solomon.

Response:

We thank the referee for this great question. According to the knots and links theory (shown in Fig. R25 below), the polar Solomon rings can be formed only in the rhombohedral phase ferroelectric because it possesses 8 variants, and could form two crossed vortices in 3D space with four crossing points, as shown in Fig. R26 below. According to the recent work, the tetragonal phase ferroelectric, which owns 6 variants, has been reported to show a hopfion structure (Fig. R27 below), which means that any two of the links interact with each other with two crossing points [15]. We have added the above discussion in the revised manuscript (page 4, paragraph 1).

Fig. R25. Stereoscopic representations of multiple knots and links. **a**, Unknot, 0_1 . **b**, Trefoil knot, 3_1 . **c**, Figure-eight knot, 4_1 . **d**, 5_1 . **e**, 5_2 . **f**, Two-component trivial link, 0_1^2 . **g**, Hopf link, 2_1^2 . **h**, Solomon link, 4_1^2 . **i**, Whitehead link, 5_1^2 . **j**, Star of David link, 6_1^2 .

Fig. R26. **a**, **b**, A 3D polar vortex composed of $R_4^-[111]$, $R_3^+[111]$, $R_2^+[111]$, $R_1^+[111]$, and $R_4^+[111]$, $R_3^-[111]$, $R_2^-[111]$, $R_1^-[111]$ polarization variants in rhombohedral phase BiFeO_3 . **c**, Distorted 3D polar Solomon rings which can be considered as two intersecting vortices composed of eight polarization variants of BiFeO_3 as shown in (a) and (b). **d**, The definition of polar Solomon rings with the linking number (LN) of +2, from the mathematical point of view.

Fig. R27. Formation of the Hopfion in a tetragonal ferroelectric nanoparticle. **a**, Uniform distribution of the polarization, P, (green lines) in a spherical nanoparticle,

blue arrows indicate the direction of polarization. Positive and negative depolarization charges on the surface induce a depolarization electric field, E_d (red lines). **b**, Polarization vortex. **c**, Escape of the polarization vortex into the third dimension. **d**, Polarization Hopfion.

References:

[15]. Hopfions emerge in ferroelectrics. *Nat Commun* **11**, 2433 (2020).

5. Regarding the phase field simulation, the authors stated that “the simulation was performed with an eight-domain initial structure, ...”. I guess the eight-domain initial structure is close to the polar Solomon state, such that it is easy to get the polar Solomon as a stable state. However, is it possible to stabilize the Solomon state in the phase field simulation by starting from a random distribution of the polarization field?

Response:

We thank the referee for pointing out this point.

Yes, the Solomon rings can also develop from a random domain. As shown in Fig. R28 below, a random domain is transformed into quad-domains after an out-of-plane electric field of 8 MV/m is applied, which is further transformed into Solomon rings after a reversed electric field of -0.5 MV/m is applied to the nanocrystal.

Fig. R28. Domain evolution from a random state to Solomon states. After 1000 steps, an out-of-plane electric field of 8 MV/m is applied to the nanocrystal. After 4000 steps, a reversed electric field of -0.5 MV/m is applied to the nanocrystal.

REVIEWERS' COMMENTS

Reviewer #1 (Remarks to the Author):

The authors have addressed my concerns properly.
This manuscript can be accepted for publication in Nature Communications in the present form.

Reviewer #3 (Remarks to the Author):

The authors have addressed all my comments in details, I would like to recommend for publication in Nature Communications.

A Point-by-Point Response Letter (for ID: NCOMMS-22-52845A)

We greatly appreciate the editor and reviewers for their in-depth review of our manuscript once again, especially for their recommendation for publication our manuscript in Nature Communications.

Reviewer #1 (Remarks to the Author):

The authors have addressed my concerns properly.
This manuscript can be accepted for publication in Nature Communications in the present form.

Response:

We thank Reviewer #1 for the suggestion of acceptance of our work.

Reviewer #3 (Remarks to the Author):

The authors have addressed all my comments in details, I would like to recommend for publication in Nature Communications.

Response:

We thank Reviewer #3 for the recommendation for publication our manuscript in Nature Communications.